# Low Vitamin B12 and Lipid Metabolism: Evidence from Pre-Clinical and Clinical Studies

**DOI:** 10.3390/nu12071925

**Published:** 2020-06-29

**Authors:** Joseph Boachie, Antonysunil Adaikalakoteswari, Jinous Samavat, Ponnusamy Saravanan

**Affiliations:** 1Division of Metabolic and Vascular Health, Warwick Medical School, University of Warwick, Coventry CV2 2 DX, UK; J.Boachie@warwick.ac.uk (J.B.); J.Samavat@warwick.ac.uk (J.S.); 2Department of Biosciences, School of Science and Technology, Nottingham Trent University, Nottingham NG11 8 NS, UK; 3Diabetes Centre, George Eliot Hospital NHS Trust, College Street, Nuneaton CV10 7 DJ, UK

**Keywords:** vitamin B12 (B12), lipid metabolism, cardiovascular disease (CVD), obesity, metabolic syndrome (MetS), type 2 diabetes mellitus (T2D)

## Abstract

Obesity is a worldwide epidemic responsible for 5% of global mortality. The risks of developing other key metabolic disorders like diabetes, hypertension and cardiovascular diseases (CVDs) are increased by obesity, causing a great public health concern. A series of epidemiological studies and animal models have demonstrated a relationship between the importance of vitamin B12 (B12) and various components of metabolic syndrome. High prevalence of low B12 levels has been shown in European (27%) and South Indian (32%) patients with type 2 diabetes (T2D). A longitudinal prospective study in pregnant women has shown that low B12 status could independently predict the development of T2D five years after delivery. Likewise, children born to mothers with low B12 levels may have excess fat accumulation which in turn can result in higher insulin resistance and risk of T2D and/or CVD in adulthood. However, the independent role of B12 on lipid metabolism, a key risk factor for cardiometabolic disorders, has not been explored to a larger extent. In this review, we provide evidence from pre-clinical and clinical studies on the role of low B12 status on lipid metabolism and insights on the possible epigenetic mechanisms including DNA methylation, micro-RNA and histone modifications. Although, there are only a few association studies of B12 on epigenetic mechanisms, novel approaches to understand the functional changes caused by these epigenetic markers are warranted.

## 1. Introduction

Obesity is currently a worldwide epidemic that results in higher insulin resistance and increases the risk of developing metabolic disorders like type 2 diabetes mellitus (T2D), hypertension and cardiovascular diseases (CVDs) [1], thus, posing a critical public health concern [2]. Approximately 2.1 billion (30%) humans worldwide are reported to be obese or overweight, contributing to 5% of global mortality. If sustained, the prevalence rate of obesity is likely to increase to 50% of the global adult population by 2030 [3]. In addition, studies from the global hunger index have reported that, approximately 2 billion people globally are affected by deficiency of micronutrients. The world health organization (WHO) is specifically concerned about the levels of vitamin B12 (B12) and folate (B9) due to increasing prevalence of their deficiencies across the populations [4]. Epidemiological studies have also clearly shown the association of these nutritional metabolites and manifestations of metabolic risk [5,6,7,8].

Dyslipidemia is a key risk factor for atherosclerosis and CVD. Studies have also shown the association of low B12 with obesity, hypertension, T2D and metabolic syndrome (MetS) in diverse populations. Low B12 may also be associated with adverse lipid profile and CVDs [9]. In pre-clinical studies, low B12 levels might increase lipid accumulation in adipocytes and trigger dyslipidemia in mice [10], suggesting that low B12 and dyslipidemia might be causally related.

Obesity could be developed principally, as a result of excessive macronutrient intake and/or reduced energy expenditure, which in-turn triggers disruption in lipid and glucose homeostasis. However, the contribution of low levels of micronutrients, such as B12 to the pathogenesis of obesity and dyslipidemia has not been fully explored. This review is aimed at summarizing the current knowledge and latest evidence of the effect of low B12 on lipid metabolism, particularly the clinical studies and epidemiological observations from pregnant women, adolescents, and adults. It will also summarize the pre-clinical evidence from cell-lines, animal models and the possible molecular and epigenetic mechanisms involved.

The current review has been produced following a detailed search of literature using the MEDLINE/PubMed [11] and Global Health (CABI) [12] databases. Medical subject headings (MeSH) and keywords such as “vitamin B12 deficiency”, “insufficiency of B12”, “vitamin B12” or “cobalamin” were used. Combinations of these keywords with “pregnancy”, “vegetarians”, “lipid metabolism” and “metabolic disorders” were achieved using Boolean operators (OR, AND).

## 2. B12: Biochemical Structure, Sources, Bioavailability, Cellular Uptake and Metabolism

B12 (cobalamin) is classified as an essential vitamin as it is entirely obtained from diet. It is also synthesized naturally by some large intestine-resident bacteria in humans [13]. However, the site of absorption in the small intestine is significantly distant from the site of synthesis, accounting for non-bioavailability of the naturally synthesized B12 in humans [14,15]. B12 was first isolated in the year 1948 [16], following the discovery of using liver extract as a source of therapy for pernicious anemia [17]. Using x-ray crystallography, B12 was structurally described as a massive organometallic compound with its size ranging between 1300 to 1500 Da [18]. The vitamin is uniquely composed of a central cobalt atom linked to six ligands, with four of the latter structurally reduced to form a corrin ring that connects and encircles the former through direct nitrogen linkages. Directly below the central cobalt is an α-axial 5,6-dimethylbenzimidazole (DMB) ligand which, through a phosphoribosyl moiety, links to the corrin ring and confers a high specificity on the vitamin for intrinsic factor (IF) binding in the lower gastrointestinal tract [19]. The β-axial ligand (R-ligand), positioned above the corrin ring, may differ in diverse forms such as methyl, 5′–deoxyadenosyl, hydroxo, acquo or cyano groups. These are, therefore, named as methylcobalamin, deoxyadenosylcobalamin, hydroxycobalamin, acquocobalamin and cyanocobalamin, respectively.

B12 is predominantly absent in plants, however, it is reported that traces of B12 could be obtained in dried purple and green lavers and some edible algae [20]. Natural sources are predominately present in animals and animal products such as meat, eggs and milk. The loss of fat and moisture associated with cooking of beef has been reported to account for an average loss of 27–C33% B12 per unit nitrogen, suggesting that extensive cooking might decrease the levels of B12 in food [21]. The bioavailability of B12 in eggs is relatively lower (<9%) compared with other sources such as meats from chicken (61–66%), sheep (56–89%) and fish (42%) [20]. It is also presumed that in healthy adults, with non-defective gastrointestinal tract (GIT), the bioavailability of dietary B12 is about 50%. In addition, the GIT-resident microorganisms possess the enzymes needed for B12 biosynthesis [22,23]. The absorptive capability of the GIT of an individual undergoes adverse alteration with age, affecting the bioavailability of B12 in humans [24]. However, the absorption of B12, aided by availability of calcium, occurs at terminal ileum where precise receptors (cubam) are expressed on the microvilli on the intestinal endothelium. These receptors bind the IF-B12 complex followed by B12 delivery into the peripheral blood [24,25].

In humans, the plasma proteins transcobalamin (TC) and haptocorrin (HC) are proposed to be crucial in the transport of B12 through circulation. TC, demonstrating a relatively increased selectivity for B12 in humans, forms a TC–B12 complex called holo-transcobalamin (holo-TC) which is reported as the biologically active proportion of B12 within circulation [26]. The precise role of HCs in humans is unclear, although they bind the greater proportion (80%) of circulating B12. HCs are reported to express a relatively lower selectivity for B12, compared to TC, suggesting that they might either play a role in the clearance of damaged forms of B12 or serve as B12 reservoirs in humans [26]. Generally, within all body cells including hepatocytes (Figure 1), B12 is believed to undergo processing into key cytosolic (MeCbl) and mitochondrial (AdoCbl) co-enzyme forms. A significant amount of the biologically active water soluble B12 is believed to accumulate in the liver of healthy humans. In liver biopsy samples from healthy humans, the average B12 content per gram of wet liver tissue is 1.94 µg (range: 1.41 to 2.58 µg) [27]. It is reported, however, that B12 content of the human liver may be lower in cases of cirrhosis, viral hepatitis, fatty liver and or obstructive jaundice [27]. Moreover, the human body is known to eliminate a total of about 2 to 5 µg of B12 daily, primarily, via feces and minimal amounts in urine [28].

## 3. Vitamin B12 Deficiency: Diagnostic Markers and Risk Factors

There is no international consensus on the lower limit of B12 to define B12 deficiency in adults and pregnancy. But there is some agreement that it should be between 120 and 220 pmol/L (higher threshold in pregnancy) with the upper limit between 650 and 850 pmol/L [29]. Other parameters, such as holo-TC, methyl malonic acid (MMA) and homocysteine (tHcy), are better markers of tissue level B12 deficiency, especially when the serum B12 levels are borderline (150–220 pmol/L) [5]. In the expanded newborn screening, estimation of metabolites such as propionylcarnitine, MMA and tHcy in samples of newborn dry blood spot, using the tandem mass spectrometry, can facilitate the early diagnosis of infant B12 deficiency resulting from maternal low B12 [30]. Since elevation of the metabolites MMA and tHcy is associated with congenital disorders of B12 (as shown in Table 1), these also should be considered in the differential diagnoses [31].

Currently, there is limited data on the prevalence of B12 deficiency globally, based on tHcy MMA levels [32]. The risk of developing B12 deficiency is higher in vegetarian populations such as in India [33]. However, it is not uncommon in other populations, ranging between 10 and 30% [29]. And is higher in pregnant populations [29]. Some of the causes of B12 deficiency are shown in Table 1.

## 4. Association of Low B12 with Cardiometabolic Risk

### 4.1. Evidence from Pre-Clinical Studies

Currently, pre-clinical evidence shows that the effect of low B12 on cardiometabolic risk has been demonstrated mainly using animal models (Table 2). Emerging studies have begun testing some of these hypotheses on some human cell line and primary cell models. Potential mechanisms underlying the low B12-cardiometabolic risk associations are being elucidated at the pre-clinical level. In animal studies, low or deficient B12 level was defined based on plasma levels of B12 in animals fed with specified diet whereas in in vitro studies, based on the levels of B12 present in the culture media. The pre-clinical evidence of low B12 that have a role on cardiometabolic risk are provided below.

#### 4.1.1. Vitamin B12 and Obesity

A study in adult Wistar rats that were fed a B12 restricted diet (control−0.010 mg/kg B12 vs. B12 restricted–0.006 mg/kg B12) during maternal or postnatal period, predicted higher visceral adiposity and resulted in alteration in the metabolism of lipids in the offspring [35]. Low plasma B12 (277 pg/mL B12) resulting from B12-restricted diet or a combination of B12- and B9-restricted diet (219 pg/mL B12) for a period of three months in pre-pregnant Wistar rats resulted in increased body weight compared to control (1164 pg/mL B12) [35]. B12-deficient rats had increased total body fat whereas B9-deficient rats presented with elevation in visceral fat mass [35]. Likewise, in female C57 BL/6 mice, severe decrease in plasma B12 (138 pg/mL), but not moderate (208 pg/mL) (compared to 406 pg/mL control B12) was reported to induce higher adiposity and altered lipid profile in their offspring [10].

#### 4.1.2. Vitamin B12 and Insulin Resistance

Restriction of maternal B12/folate/methionine (control vs. restricted: B12 (1000.5 pM vs. 198 pM), folate (6.90 nM vs. 4.42 nM), methionine (39.1 µM vs. 30.8 µM)) at the stage of conception in sheep models, showed increased resistance to insulin and elevated blood pressure in its offspring [36]. In addition, the adult male offspring had higher adiposity and altered functioning of their immunity. This evidence was explained by the observation of male-specific demethylation of the affected loci, therefore, providing convincing reasons for the these differences observed in the phenotypes of the offspring [36]. Finally, the study concluded that the decreased methylation of DNA could be explained by lower availability of s-adenosyl methionine (SAM), a well-known methyl donor, which is crucial in epigenetic modulations underlying the development of resistance to insulin [36,37].

#### 4.1.3. Vitamin B12 and Dyslipidemia

Mother Wistar rats with low plasma B12 (277 pg/mL B12 compared to 1164 pg/mL control) produced offspring which had increased levels of adiposity, triglycerides (TG) and total cholesterol as well as decreased leptin and adiponectin, compared to control offspring, showing a dysregulated metabolism of lipids [35]. C57 BL/6 mice with deficient plasma B12 levels at 12 weeks (145 pg/mL vs. 548 pg/mL control) and 36 weeks (123 pg/mL vs. 522 pg/mL control) had increased plasma TG, cholesterol and pro-inflammatory markers comprising tumor necrosis factor-alpha (TNFα), interleukin−1 b, interleukin−6, as well as lower adiponectin concentrations [38]. B12 deficiency inhibited beta oxidation of fatty acids and lipolysis in hepatic tissues of rat offspring born to B12 deficient mothers (227 pg/mL B12 vs. 1164 gp/mL control) [39]. However, in these studies, B12 supplementation during parturition resulted in the restoration of both pathways [39]. In our recent study, targeting the elucidation of the cellular mechanism induced by low B12 in human adipocyte cell line (Chub-S7 (0.15 nM compared to 500 nM control)), elevated levels of cholesterol [40] and TG [41] were observed in low B12 cells compared with controls. This was explained by a reduction in the methylation potential and the SAM: SAH ratio in low B12 conditions [40]. Validation of the above findings was further endorsed in primary human adipocyte models, demonstrating that B12-deficient primary human adipocytes had significantly elevated levels of total cholesterol, tHcy and mRNA expression of key genes that regulate the biosynthesis cholesterol, as compared to controls [40].

#### 4.1.4. Vitamin B12 and Cardiovascular Diseases

Adult male Sprague–Dawley rat model treated with testosterone enanthate (0.5 mg/100 gm) and B12 (500 µg/kg) demonstrated significant changes with peripheral cortisol and association with vascular dysfunction [42]. Another study involving four-weeks simultaneous therapy with 10 mg/kg folate and 500 µg/kg B12 in Wistar rats, aiming to assess the effect of B12 and folate supplementation on myocardial infarction (MI) in rats presenting with hyperhomocysteinemia, there was a significant reduction in the elevated heart rate and blood pressure as well as attenuation of severe cardiac histopathological alterations [43]. The study suggested that aggravation of MI risk might result from hyperhomocysteinemia, however, B12 and/or folate administration may reduce MI risk and tHcy levels, accounting for reduced harmful consequences associated with hyperhomocysteinemia [43].

### 4.2. Evidence from Clinical Studies

Most observational and epidemiological evidence on the effects of B12 on metabolic risk are from an Asian population but there are few studies in Western populations. As previously stated, the term “low B12” in all clinical evidence reported in this review refers to plasma/serum B12 levels ≤120–300 pmol/L (Table 3).

#### 4.2.1. Vitamin B12 and Obesity

An association between maternal body mass index (BMI) at early stage pregnancy and plasma B12 and/or B9, in a recent cross-sectional study, was observed in obese women compared to women with normal BMI [44]. Sukumar et al. reported that pregnant women presenting with low B12 at the first trimester had increased BMI compared with those with normal B12 levels [45]. In India, low maternal level of B12 was suggestive of contributing to an increased likelihood of developing higher adiposity and insulin resistance in the offspring [46]. Likewise, the mean B12 concentration was significantly lesser in children with obesity, compared to healthy volunteers, and was negatively associated with the severity of obesity [47]. In a recent systematic review of clinical data, although not conclusive, lower concentrations of B12 were observed in obese individuals compared to overweight and normal weight individuals [48].

#### 4.2.2. Vitamin B12 and Insulin Resistance

In White Caucasian pregnant women without gestational diabetes mellitus (GDM), an inverse relationship between B12 level and insulin resistance was reported [49]. Low level of B12 in the plasma of mothers at early but not late pregnancy, was linked with significant increase in the resistance to insulin expressed by homeostasis model of insulin resistance (HOMA-IR) in their children [50]. However, there was no relationship established between B9 status in the mothers and insulin resistance in their children [50]. Similarly, within school children of Nepal, high insulin resistance was observed as a result of low maternal levels of B12 [51]. In obese adolescents with low or borderline levels of B12, there was an association between low B12 status and insulin resistance and obesity [52]. In support of these clinical studies, a study from our group in low B12 pregnant women demonstrated that altered circulating micro-RNAs derived from adipose tissues could possibly mediate adipogenic and insulin-resistant phenotypes as a precursor to obesity [41]. In addition, studies from other population groups such as women with polycystic ovarian syndrome, adults (males and females) in a primary care setting, and obese adolescents (males and females) also showed low B12 levels were associated with higher insulin resistance [29,53,54].

#### 4.2.3. Vitamin B12 and Dyslipidemia

Clinical studies involving three independent cohorts of women; (i) at child-bearing age, (ii) in early pregnancy, and (iii) at delivery, revealed that low levels of B12 were associated with elevated levels of low-density lipoprotein (LDL)-cholesterol, total-cholesterol and cholesterol-to-high-density lipoprotein (HDL) ratio [40]. The study also showed that the women had high prevalence of low B12 (14% in child bearing age, 45% in early pregnancy, 40% during delivery) [40]. Maternal subcutaneous adipose tissues (ScAT) obtained from mothers with low circulating B12 showed upregulation of genes involved in the biosynthesis of cholesterol, such as 3-hydroxy−3-methylglutaryl-CoA reductase (HMGCR), sterol regulatory element binding protein 2 (SREBF2), low-density lipoprotein receptor (LDLR) and sterol regulatory element binding protein 1 (SREBF1) [40]. These studies implicate that low B12 status might be causally linked to increased levels of low-density lipoprotein cholesterol (LDL-C), total cholesterol, cholesterol-to-high-density lipoprotein (HDL) ratio and subsequent insulin resistance. Further evidence from other studies have shown that babies born to low B12 mothers are ‘thin-fat’, a term describing phenotypically lean individuals with increased fat accumulation in their bodies as well as decreased lean mass [46,55]. This may consequently lead to elevated resistance to insulin as well as increased risk of developing CVD in adulthood [46,55]. In addition, studies in patients with non-alcoholic fatty liver disease (NAFLD) and non-alcoholic steatohepatitis (NASH) showed that low B12 level significantly increased the levels of triglycerides, cholesterol and blood glucose levels in these patients [56]. A study in adult population from North India showed that low B12 level was observed to be significantly associated with both low HDL and hyperhomocysteinemia, whereas after controlling for the confounder tHcy, B12 was found to be associated with all lipid indices including HDL [57]. A negative correlation was also observed between B12 levels and the prevalence of MetS in euthyroid participants [58].

#### 4.2.4. Vitamin B12 and Cardiovascular Diseases

B12 plays a crucial role in the generation of methionine from tHcy via re-methylation [59]. Therefore, low B12 is associated with an elevation of circulatory tHcy which is known to associate with the risk of CVD [60]. In Chinese subjects, a study showed a correlation between the highest circulating levels of B12 and the lowest tHcy levels in patients presenting with CVD [61]. Similarly, in Japanese subjects, evidence of an inverse relationship was observed between dietary folate (B9) and B6 intake and the incidence of heart failure mortality in men as well as mortality resulting from stroke, coronary artery disease (CAD) and total CVD in women [62]. This was similar to the evidence in an American population study in which therapy for about 7.3 years with a B12–B6–folate combination pill resulted in a significant decrease in tHcy levels [63]. Other studies suggested that supplementation with 250 µg B12 and 5 mg folate resulted in 32% reduction of fasting plasma tHcy levels following 12 weeks of therapy in patients with CAD [64]. A study by Setola et al. [65] demonstrated that folate and B12 treatment in patients with MetS improved resistance to insulin and endothelial dysfunction as well as decreased tHcy levels, suggesting the beneficial effects of these vitamins on CVD risk factors. Recent evidence has shown that elevated tHcy and decreased circulatory B12 levels in women, demonstrated a strong association with higher risk of all-cause and CVD deaths in the elderly population [66]. Meta-analysis of several prospective studies shows reliable evidence of a correlation existing between plasma tHcy and elevated CVD risk [67]. For instance, in different randomized control trials (RCTs) involving an established renal disease or CVD, B12 supplementation using a dosage range of 0.4–1.0 mg B12 per day accounted for a significant reduction in the risk of developing stroke [67]. Meanwhile, studies have proposed that hyperhomocysteinemia and cardiovascular risk in patients may precede the development of end-stage renal disease (ESRD), chronic kidney disease (CKD) and dialysis [68,69,70]. Although CKD patients demonstrate an impairment in tissue uptake of B12 resulting in functional deficiency [71], the current evidence remains unclear to consider altered B12, folate and elevated tHcy levels as markers for CVD and cardiovascular mortality risk in ESRD and CKD individuals [72].

## 5. Molecular Mechanisms Regulating Low B12 and Lipogenesis

Environmental factors such as nutrition may affect lipid metabolism by targeting at the transcriptional level, thereby, regulating the expression of genes and subsequently the phenotype without altering the sequence of nucleotides by the process of epigenetics [73]. B12 plays a crucial role in humans by serving as a cofactor in the reaction pathways essential for maturation and preservation of cells [74,75]. In the cytosol, B12 is a cofactor for methionine synthase (MS) (Figure 2), which is actively involved in the biosynthesis of methionine via re-methylation of tHcy and in turn generation of s-adenosyl methionine (SAM), a potent donor of methyl groups for methylation of several biochemical processes. Currently, the well-known markers of epigenetic mechanisms that regulate the metabolism of lipids and resistance to insulin are DNA methylation, microRNAs, chromatin remodeling as well as modification of histones [76,77]. B12 acting as a co-enzyme for a source of methyl groups (SAM), regulates these epigenetic mechanisms and several transcriptional as well as post-translational factors involved in the process of de novo lipogenesis. In the mitochondrial based propionate metabolism pathway, B12 serves as a cofactor in the conversion of methyl malonyl-CoA to succinyl-CoA by methyl malonyl-CoA mutase (MCM) (Figure 2) [78]. Succinyl-CoA is utilized as substrate in the Krebs cycle for ATP synthesis essential for the sustenance of cellular metabolisms (reviewed by [79]) and utilized for hemoglobin synthesis in red blood cell production. Low B12 levels, therefore, results in a reversible increase in methyl malonyl-CoA leading to accumulation of MMA. MMA, in turn, acts as a potent inhibitor of the rate limiting enzyme carnitine palmitoyl transferase 1 (CPT1), critical for the breakdown of long chain fatty acids in the beta oxidation pathway. This inhibition of CPT1 may build up fatty acids and TGs (reviewed by [80]) accounting for higher lipogenesis and insulin resistance [81] which might result in dyslipidemia.

## 6. Epigenetic Mechanisms Underlying Low B12 Levels in Lipid Metabolism

There is currently, emerging evidence suggesting that abnormalities associated with one-carbon metabolites due to low B12, might possibly exert their modulations through epigenetic mechanisms. Some preliminary findings have indicated that induction of alterations in normal methylation of DNA is due to changes in the levels of one-carbon metabolites. However, it was suggested that in order to elucidate the molecular mechanism underlying these effects, there is a need for future studies to target combinations of gene expression assays with epigenetic studies [79]. Studies in animals are likely to be more informative as clear elucidation of molecular mechanisms in humans might be difficult due to tissue-specificity of the epigenetic phenomena [79].

### 6.1. DNA Methylation and Regulation of Lipid Metabolism through Low B12

Methylation generally involves the process of transferring a methyl group to substrates such as enzymes, proteins, amino acids and DNA in diverse cells and tissues [82]. In the process, guanine is normally bound with 5-methylcytosine following its development from cytosine [83]. Usually, hypermethylation of global DNA and 5′-cytosine-phosphate-guanine−3′ (CpG) islands located within the promoters of genes involved in lipid synthesis, accounts for the stability of the genome and silencing of the lipogenic genes, respectively [84]. This is greatly dependent on the availability and sufficiency of SAM synthesized with the aid of methyl donors such as B12 and folate [76]. Low B12 independently impairs the MS action [85] affecting SAM/SAH ratio which is a principal determinant of the methylation potential of cellular DNA, thereby inducing dysregulation of gene expressions [86,87]. Demethylation associated with DNA of the genome, persists as one of the earliest evidences of epigenetic modifications triggered by diets deficient in methyl donors, contributing to hepatocarcinogenesis in animals [88].

A study in zebrafish showed that micronutrient-deficient diet (without 1-carbon nutrients B12, B9, B6, methionine and choline supplementation compared to control), fed to parents affected the hepatic methylation of DNA and gene expression of lipogenesis leading to lipid accumulation in the F1 offspring [87]. High global methylation of DNA and promoter region methylation of lipid genes are critical to the maintenance of normal physiology of metabolism [89,90] by ensuring stability of the genome and silencing of abnormal lipogenic genes [84]. A study in an adipocyte model showed that hypomethylation in promoter regions led to higher expressions of LDLR and SREBF1 genes and cholesterol biosynthesis in low B12 (0.15 nM B12 media compared to 500 nM control) [40]. Increased genome incorporation with uracil subsequent to hypomethylation was observed in B12-deficient rats (50 µg/kg B12 fed to Sprague–Dawley rats) [91], whereas fatty acid supplementation in low B12 conditions could not restore global methylation [92]. Evidence of hypomethylation, as seen in low B12, was observed in the CpG sites of some liver-derived genes involved in the pathogenesis of T2D due to low folate levels [93]. Methyl donor supplementation in rat obesogenic models ameliorated hypomethylation near the promoter of genes such as acylglycerol−3-phosphate-O-acyltransferases 3 (AGPAT3), SREBF2 and ESR1 (estrogen receptor−1) resulting in the reversion of higher fatty acid, TG and cholesterol accumulation in the liver [94]. These studies thus implicate that methyl donor deficiency compromises methylation capacity, resulting in the dysregulation of lipid metabolism [87,95,96] which can be reversed by supplementation of methyl donors.

### 6.2. MicroRNAs (miRNAs) and Regulation of Lipid Metabolism through Low B12

MicroRNAs (miRNAs), formerly termed as small temporal RNAs (stRNAs), are the class of small RNAs comprising 21–25 nucleotides of single-stranded RNA which are highly conserved and engaged in regulating the expression of genes at the transcriptional and post-translational levels [97]. MiRNAs physiologically express diversified patterns and engage in the regulation of certain genes underlying the metabolism of lipids and inflammation [98,99]. MiRNAs are proposed to be engaged in the modulation of adipocyte differentiation accounting for development of insulin resistance, T2D and dyslipidemia [100,101]. Genes regulating hepatic metabolism of fatty acids and/or insulin signaling may be repressed by some miRNAs regulating the levels of HDL, TG and insulin [102]. About 100 different miRNAs are expressed differentially in cases of NASH in humans [103] and alterations in miRNA-29c, miRNA-34a, miRNA-200b and miRNA-155 are due to methyl-donor deficiencies [104]. In humans, miRNA-122 is largely expressed in hepatocytes but under expressed in NASH [104] and therefore is proposed to be the potential target for treatment of dyslipidemia [105] and high cholesterol [106]. Using adipocyte models, our group reported that twelve different adipocyte-derived miRNAs targeting peroxisome proliferator-activated receptor gamma (PPARγ) (miRNA-31, miRNA-130b and miRNA-23a), adipocyte differentiation (miRNA-143, miRNA-145, miRNA-146a, miRNA-125b, miRNA-222 and miRNA-221), CCAAT/enhancer-binding protein alpha (miRNA-31) and pathways of insulin resistance (miRNA-107 and miRNA-103a) were significantly altered due to low B12, thereby modulating adipocyte differentiation and physiology [41]. In addition, B12 levels were positively associated with seven different circulating miRNAs (miRNA-27b, miRNA-130, miRNA-103a, miRNA-107, miRNA-125b, miRNA-23a, miRNA-221 and miRNA-222). Of these, four (miRNA-107, miRNA-27b, miRNA-23a and miRNA103a) were inversely and independently associated with maternal BMI, similar to low B12 after adjusting for likely confounders (such as age, parity, smoking, supplement use, glucose and insulin). The reduction in effect size between B12 and BMI on additional adjustment for these four miRNAs, highlights that some of the negative effects of B12 on BMI may be mediated by these miRNAs [41].

### 6.3. Modifications of Histones and Regulation of Lipid Metabolism through Low B12

The mechanistic effect of B12 on hepatic lipid metabolism regulation via modulation of histones is primitive. However, mechanisms including methylation, acetylation, phosphorylation as well as ubiquitylation are proposed to be involved in ubiquitylation modification of histones that contribute to lipid metabolism regulation in the liver. The most intensively understood mechanism directly involved in dysregulation of hepatic lipid metabolism is histone acetylation which is normally catalyzed by histone acetyltransferases (HATS) and histone deacetylases (HDACs). In female C57 BL/6 mice, chronic deficiency of B12 (580 pmol/L B12 compared to 2331 pmol/L control) showed a dysregulated histone-modifying enzyme expression in the brain and abnormal behavioral anomalies [107]. Recent evidence from pilot data indicated that low folate level in the liver was associated with higher levels of methylation in H3 K4 [108]. Reduced methylation of histone has also been shown in liver X receptor-alpha (LXRα) in rats, following consumption of a high-fat diet after the third generation in rats [109]. Histones such as H3 lysine 4 (H3 K4) and H3 lysine 9 (H2 K9) are known to be subjected to regulation by methylation of the histones in association with DNA methylation [110]. Evidence of deficiency in the methyl donor choline in C57 BL/6 mice at the 12th–17th day of gestation resulted in alterations in the methylation of histone H3, whereas choline after undergoing conversion to betaine, enhanced re-methylation to tHcy [111]. Convincing evidence shows the link between NAFLD [73], resistance to insulin and obesity with modification of histone such as demethylation of H3 at lys9 [112]. Conditions of hyperlipidemia and obesity were observed in mouse models following the functional loss of Jhdm2 a, the H3 K9-specific demethylase, which implies that the status of methylation of H3 K9 is very essential and may tend to regulate the expression of metabolic genes [113]. Further studies are recommended to determine the exact role of B12 on the metabolism of lipids via modulation of histones.

## 7. Impact of One-Carbon Micronutrient Supplementation on Lipid Metabolism

In recent times, although agents that potentially resolve insulin resistance appear promising, the general therapy for correcting dysregulation of lipogenesis has not been established. However, current treatment may be reliant on alterations in lifestyle such as diet, weight reduction and/or exercise [114]. It is generally suggested that interventions involving fortification of foods, supplementation with B12 during preconception period and provision of adequate education could collectively be the most effective ways to enhance improvement in the levels of B12 in infants as well as mothers [115]. Supplementing with a diet rich in one-carbon nutrients such as vitamin B6, B9 and B12, is the principal way to significantly influence methylation of DNA because of their contribution to the synthesis of SAM. A similar micronutrient, betaine, obtained from food functions as a methyl donor and has been identified as a potent alleviator of fatty liver. Betaine was shown to facilitate the export of triglycerides from the liver as a way of attenuating steatosis, especially in cases of NAFLD [116]. Post-supplementation with only B12 was associated with significant alteration in the methylation of DNA in which 589 CpGs demonstrated differential methylation in addition to 2892 regions. Supplementation with both B12 and folate resulted in differential methylation of 169 CpGs and 3241 regions, thereby influencing the expression of genes associated with T2D [117]. Evidence of higher percentage of transmethylation to methionine and reduced transculturation to cysteine were observed as a result of supplementation with both B12 and milk protein compared with only milk supplementation in women. Therefore, among Indian women presenting with low B12 in early pregnancy, supplementation with B12 and energy-protein balance plays a crucial role in enhancing the optimum function of the methionine cycle especially at the third trimester of pregnancy [118]. Similarly, B12, folate, choline and betaine-rich methyl donor supplements were used to regress the accumulation of fats in the liver induced by high-fat-sucrose probably by changing the levels of methylation of CpG areas in the promotor regions of SREBF2, estrogen receptor 1 (ESR1) and/or acylglycerol−3-Oacyltransferase−3 (AGPAT3) [96]. Hypermethylation of DNA in fatty acid synthase (FAS) gene expression was estimated after supplementation with methyl donors that apparently boosted the retrogression of NAFLD-induced by high fat diet [94]. Normalization of histochemical indices in hepatic lobules was achieved in an animal-model study where the accumulation of lipids towards the center of hepatic lobules was brought under control subsequent to treatment with methionine in combination with vitamin B12 rather than B15 [119]. Finally, oral B12 supplementation during pregnancy resulted in higher plasma B12 levels in infants and reduction in MMA and tHcy levels [120].

## 8. Conclusions and Prospects

In summary, the global incidence of metabolic disorders is increasing with advancement in urbanization in addition to several environmental and genetic factors. The pathogenesis of MetS, T2D and CVD have been associated with deficiencies in micronutrients. The effect of low B12 on the pathogenesis of several metabolic disorders such as obesity, insulin resistance, T2D and CVD, has been studied at the pre-clinical and clinical levels. Clinically, low B12 status in children, adolescents and pregnant mothers was associated with higher adiposity and lipids, as well as increased risk of insulin resistance, T2D and CVD. Babies born to low B12 mothers via adverse maternal programming seem to have a high accumulation of adiposity and insulin levels at birth which may predispose them to a higher risk of developing cardiometabolic disorders in later life. This review provides compelling evidence that the regulation of lipids and increased adiposity are associated with low B12 via epigenetic mechanisms. Future studies with a critical emphasis to establish causality by understanding the functional relevance of these epigenetic changes induced by low B12 are required.

## Figures and Tables

**Figure 1 nutrients-12-01925-f001:**
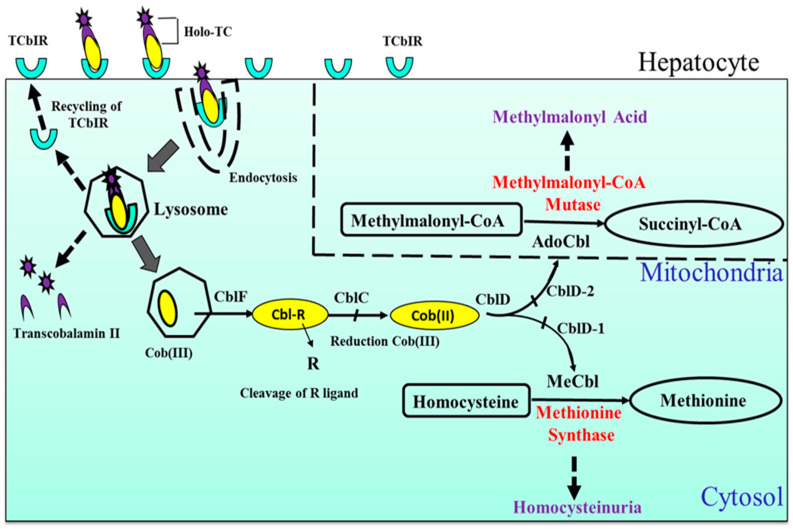
Cellular (hepatocyte) uptake and metabolism of vitamin B12 (B12): Cells generally internalize B12-bound transcobolamin (holo-TC) with the aid of transcobalamin receptor (TCR) TCbIR/CD320 via endocytosis and fused into lysosomes. Within this organelle, B12 is liberated from the TC with the latter (Apo-TC) subjected to degradation whilst the former (B12) is transported to the cytosol and further processed to its catalytic forms, methyl-cobalamin (MeCbl) and 5′-adenosyl cobalamin (AdoCbl), acting in the cytosol and mitochondria as co-enzymes in the methyl malonyl CoA mutase (MCM) and methionine synthase (MS) pathways, respectively. The transcobalamin receptors (TCbIR) are however recycled back to the cellular surface membrane.

**Figure 2 nutrients-12-01925-f002:**
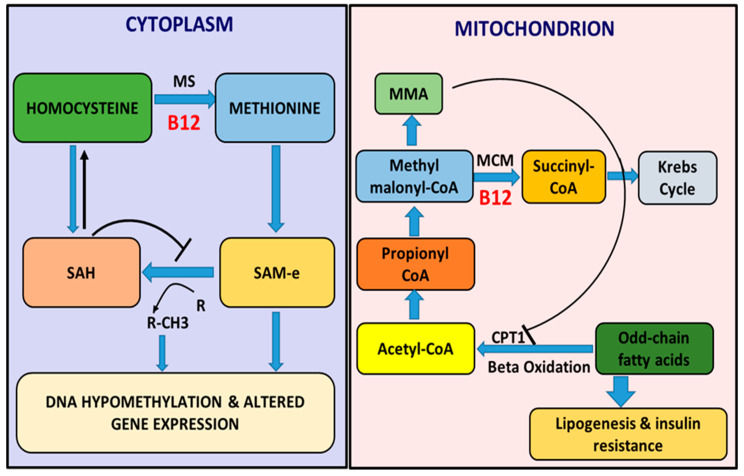
Cellular role of B12 in lipogenesis. There is a reduction in the production of methionine as well as the methyl donor s-adenosyl methionine (SAM) within the cell’s cytosol, resulting from B12 deficiency, leading to hyperhomocysteinemia as well as reversible increase in s-adenosyl homocysteine (SAH) which is known to be an inhibitor of DNA methyl transferases (DNMTs). The inhibition of DNMTs together with low levels of SAM results in hypomethylation of DNA and altered gene expressions. Beta oxidation of fatty acid is inhibited by generation of methyl malonic acid (MMA) from methyl malonyl-CoA within the mitochondria due to insufficiency of B12, a cofactor for methyl malonyl-CoA mutase (MCM) enzyme required for the biosynthesis of succinyl-CoA from methyl malonyl-CoA in the propionate metabolism pathway.

**Table 1 nutrients-12-01925-t001:** Causes of B12 deficiency [34].

(a) Reduced B12 Intake i. Malnutrition ii. Vegetarian diet iii. Alcohol abuse iv. Old age (>75 years)
(b) An Impairment of B12 Bioavailability Via Gastric Wall Damage (with a Decrease in Intrinsic Factor)
i. Total (in certain stomach cancers) or partial gastrectomy including bariatric surgery.
ii. Atrophic autoimmune gastritis (such as pernicious anemia) or other gastritis (e.g., *Helicobactor pylori*).
(c) Impairment of Absorption Via the Intestines
i. Blind-loop syndrome
ii. Overgrowth of bacteria, giardiasis and tapeworm infections
iii. Ileal resection iv. Crohn’s disease
(d) Inherited (Congenital) Disorders of B12 Deficiency
i. Defect of the intrinsic factor receptor such as in ImerslundGräsback syndrome
ii. Juvenile pernicious anemia—Congenital intrinsic factor (IF) deficiency
iii. Cobalamin mutation (C-G−1 gene)
iv. Deficiency in Transcobalamin (TC) v. Methylmalonic acidemia, homocystinuria and combined methylmalonic acidemia and homocystinuria [31]
(e) Increased B12 Requirements
i. Hemolytic anemic conditions
ii. HIV infection
(f) Drugs
i. Metformin ii. Proton pump inhibitors iii. Prolonged use of histamine receptor 2 (H_2_) blockers (especially >12 months)

**Table 2 nutrients-12-01925-t002:** The effect of low B12 on components of cardiometabolic risk in pre-clinical studies.

Obesity	Insulin Resistance	Dyslipidemia	Cardiovascular Diseases
a. Increased visceral adiposity in Wistar rats [35].	Increased resistance to insulin and elevated blood pressure in sheep [36,37]	a. Increased adiposity, TG and total cholesterol levels in Wistar rat models [35].	a. Disruption of androgen testosterone levels associated with vascular dysfunction in Sprague–Dawley rat model due to low B12 [42]
b. Higher adiposity in female C57 BL/6 mice [10].		b. Increased plasma TG, cholesterol and some pro-inflammatory markers [38]	b. Reduction in myocardial infarction (MI) risk and tHcy levels due to B12 and/ or folate supplementation in rat models [43]
c. Increased body weight in Wistar rats [35,38]		c. Increased cholesterol levels in human adipocyte cell line (Chub-S7) [40]	
d. Higher total body fat in Wistar rats [35,38]		d. Increased TG in human adipocyte cell line (Chub-S7) [41].	

**Table 3 nutrients-12-01925-t003:** The effect of low B12 on components of cardiometabolic risk in clinical studies.

Obesity	Insulin Resistance	Dyslipidemia	Cardiovascular Diseases
a. Low B12 (<150 pmol/L) was associated with increased adiposity with higher T2D risk in pregnant women [46]	b. Low B12 (<180 pmol/L [49]) was associated with increased resistance to insulin in White Caucasian pregnant women without GDM.	a. Low B12 (<148 pmol/L) was associated with elevated levels of LDL-cholesterol, total-cholesterol and cholesterol-to-HDL ratio in pregnant and non-pregnant women at childbearing age [40]	a. B12 was negatively correlated with tHcy levels in Chinese-CVD patients ≥ 65 years of age (median low B12 = 4.19 pmol/L) [61].
b. Low B12 (median−203 pmol/L [44] and <148 pmol/L [48]) levels were associated with increased BMI in maternal and general and/or clinal populations respectively	b. Prediction of higher risk of resistance to insulin in children born to low B12 (<150 pmol/L [46], 148 pmol/L [50]) mothers	b. Low B12 (≤220 pg/mL) was associated with both low HDL and hyperhomocysteinemia in North Indian population [57]	b. Low B12 (<148 pmol/L) and high tHcy levels were associated with higher risk of all-cause and CVD deaths in aged women [66]
c. Low B12 (102–208 pmol/L interquartile range) was associated with obesity in children compared to healthy volunteers [47].	c. Low B12 (≤150 pmol/L [29], <148 pmol/L [52], <200 pg/mL [53], <178 pg/mL [54]) levels were associated with increased IR in adult patients, polycystic ovarian syndrome women and obese adolescents	c. B12 negatively correlated with markers of MetS such as low HDL and high TG levels in erythroid patients (Low B12 range 180–301 pmol/L) [58]	c. B12 supplementation reduced the risk of stroke in patients with CVD and/or renal disease [67].

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
