# Peer review of "Low Vitamin B12 and Lipid Metabolism: Evidence from Pre-Clinical and Clinical Studies"

_nutrients, 2020, doi:10.3390/nu12071925_

Round 1

Reviewer 1 Report

The authors  effectively replay to the criticisms

Author Response

Many thanks

Reviewer 2 Report

ID: nutrients-832920

Boachie te al., reviewed the relationship between the content of Vitamin B12 and lipid metabolism.

I think that the review is interesting but it requires major revision listed below:

i.                   The authors refers to low B12 levels. It is necessary to define the figures of concentration of Vitamin B12. I suggest the authors introduce a table with the value of vitamin B12 levels especially the calues defined a slow

ii.                 As evidenced by several studies, it is known that the increase in propyonil carniitne and methylamolinic acid on spot analyzed in tandem mass spectrometry for the expanded newborn screening after the birth of newborns can be correlated with low levels of B12 in mother and newborn, so that it is necessary to make differential diagnosis with congenital metabolic disorders of the B12, such as METHYLMALONIC ACIDEMIA.

Authors must introduce a paragraph on newborn screening relative to maternal B12 deficit

iii.              Table 1 must be completely revised. For example:

a). reduced B12 intake can be also ascribed to alcohol abuse or age (patients older than 75 years)

b). an impairment of B12 bioavailability via gastric wall damage (with a decrese of intrinsic factor) can be due to atrophic autoimmune gastritis (such as pernicious anemia) or other gastritis (such as g. by HELICOBACTER PYLORI)

c). impairment of absorption via the intestines could be to Crohn disease, lieal resection, tapeworm infection

d). inherited conditions of B12 deficiency could be due to methylmalonic acidemia, combined methylmalonoic acidemia and homocystinuria and homocystinuria) see for example Diosrders of intracellualr Cobalain metabolsim CP Venditti 2008

f). drugs must include histamine H2 blocker use for more than 12 months

iv.              English must be carefully checked by a native speaker

Reviewer 3 Report

Great review and manuscript on B12 levels and lipid metabolism. The role of B12 deficiency on different parts of metabolic syndrome has been reported. The relation of B12 status on lipid metabolism is new and this focus may be clinically relevant. I have no specific comments regarding this manuscript. 

Author Response

Many thanks

Round 2

Reviewer 2 Report

accept

This manuscript is a resubmission of an earlier submission. The following is a list of the peer review reports and author responses from that submission.

Round 1

Reviewer 1 Report

This narrative review  try to provide evidences on vitamin B12 and metabolic disorders including lipid metabolism. 

The main conclusion is that there is no evidence whether dyslipidemia and increased adiposity due to b12 deficiency are directely linked with regulation in the metabolisms.  I think that the editing of the review should be organized according 

1) title should be more informative e.g Lack of evidence on role of ......

2) Abstract should be  modified in the last para  specifing that no evidence are reached . The last sentence should be omitted since is too speculative 

3) Authors should make a table(s)  grouping the studies on the basis of the four items investigated : vit b12 and obesity , vit b12 and insulin resistance , vit b12 and dyslipidemia and vitamin b12 and cardiovascular diseases , in order to give the posibility to the readers to easily evaluate the data.

4) Figure 1 and 2 should be omitted   and relative paragraphs merged since the informations very well know 

5)Table 1 is inaccurate : decreased iingestion of animal ric- products ,means  vegeterian ; impairtment of B12 biovailability  due gastric damage ; Add before pernicious anemia  autoimmune gastritis; omit the luminal disturbances ;  omit nitrous oxide  , write correctely proton pump inhibitors

6) In the conclusion page 12 last sentence  is highly speculative  and contaddicting the conclusion of the review

Reviewer 2 Report

The topic of this paper is of importance, mainly because there are really few revisions addressing this topic while the studies are also scarce so it is a good starting point for future research. 

As a general comment on the manuscript, sometimes is difficult to read due to the high number of abbreviations, number of genes and molecules. I´m not saying that you should give up specificity in your review but meanwhile you should try to be appealing for the reader and this you may achieve by being more systematic and pragmatic in presenting your results. Besides, it seems B12 is really important during pregnancy, pregnant women and their babies so either you specify this in the aim of this review or either you can have an special comment in the conclusión due to the scarcity of the studies on B12 in other population groups and discuss a bit.

Line 51: triggers? Typo error

Line 81: extra spaces

Point 3. You may develop more on sources. Animal vs vegetal and % of bioavailability. Also, the effect of cooking methods…

In point 3 also different line spaces

Point 4. Due to the open title that may apply to different population groups, why did you only focus on adults and pregnant women? Is there a reason for that? You may consider to include also other population groups…

Point 5: Take care with spaces between lines and between words like in line 133…

Again in this point you focus a lot on pregnant women without any special established basis from the beginning of your review. I would recommend to be more systematic in presenting the results of your review. Besides, this point is a bit unustructured and difficult to read. I highly recommend to be clearer in presenting the results maybe by splitting up the paragraphs by topics or by population groups…

Line 175: men instead of non-pregnant adults?

Line 199: recent study? Typo error

Round 2

Reviewer 1 Report

The revised version of the  manuscript  address all the previous concerns